# Sulfated Polysaccharide Isolated from the Nacre of Pearl Oyster Improves Scopolamine-Induced Memory Impairment

**DOI:** 10.3390/antiox10040505

**Published:** 2021-03-24

**Authors:** Hikaru Yamagami, Tatsuya Fuji, Mayumi Wako, Yasushi Hasegawa

**Affiliations:** College of Environmental Technology, Muroran Institute of Technology, 27-1 Mizumoto, Muroran 050-8585, Japan; 19041084@mmm.muroran-it.ac.jp (H.Y.); 10999537@mmm.muroran-it.ac.jp (T.F.); 17023128@mmm.muroran-it.ac.jp (M.W.)

**Keywords:** memory, pearl oyster nacre, polysaccharide, scopolamine, traditional medicine

## Abstract

Pearl and nacre have been used in traditional medicines for treating brain dysfunctions, such as epilepsy, myopia, palpitations and convulsions. We previously showed that a pearl oyster nacre extract improves scopolamine-induced memory impairments using the Y-maze, Banes maze and object recognition tests. In this study, we aimed to isolate the memory-improving substance using ion-exchange column chromatography and reverse-phase column chromatography and elucidate the molecular mechanism underlying its memory-improving activity. The isolated substance was found to be a sulfated polysaccharide with a molecular weight of approximately 750 kDa. Monosaccharide composition analysis showed that it was rich in galactose, glucose, mannose and uronic acid. Furthermore, the mRNA expression levels of oxidative stress, inflammatory response and neuroprotective factors in the cerebral cortex were investigated. Treatment with the polysaccharide increased the expression levels of the antioxidant enzymes Cu, Zn -superoxide dismutase (SOD) and catalase and attenuated the scopolamine-mediated upregulation of the inflammatory cytokines interleukin-1 and interleukin-6. In addition, the polysaccharide suppressed the decrease in the expression levels of brain-derived neurotrophic factor (BDNF) and nerve growth factor (NGF). These findings strongly suggest that the polysaccharide in the nacre extract mediated its antiamnesic effects by preventing oxidative stress and inflammation and increasing the expression levels of BDNF and NGF.

## 1. Introduction

Pearl oyster shells are composed of a nacreous layer and a prismatic layer. Pearls, composed of the nacreous layer, are produced by mollusks, such as pearl oysters. Pearl and nacre consist of CaCO_3_ (>90%), proteins, peptides, glycoproteins, chitin and lipids [1]. Pearl powder has been used in traditional Chinese medicine to treat brain dysfunctions, such as epilepsy and myopia [2,3,4]. Pearl powder has also been used in food supplements and cosmetics.

Pearl oysters are widely used in the pearl-producing industry. As a result, approximately 100,000 tons of pearl oyster shells are generated as industrial waste per year, and most shells containing a nacreous layer are discarded in Japan [5]. Therefore, effective utilization of pearl oyster shells is strongly desired.

Previous studies on the biological activities of pearl and nacre have shown their capacity to stimulate the formation of new bones. In vivo studies have shown that new bone is formed without causing any inflammation when nacre is implanted in the bone [6,7,8,9,10]. Furthermore, in vitro studies have shown that water-soluble components extracted from the nacre promote the differentiation of preosteoblast cells and mineralization [11,12]. Many proteins have been identified to regulate the formation of nacre. In fact, p10 and p60 proteins from nacre were found to promote the differentiation and mineralization of preosteoblast cells [13,14,15].

In addition to the induction of bone formation, nacre powder promotes skin wound healing. An in vivo study showed that powdered nacre implanted in the dermis increased collagen synthesis by stimulating dermal fibroblasts. Furthermore, water-soluble components from the nacre promote the proliferation of dermal fibroblast cells, enhance collagen secretion [16,17,18] and promote wound healing following burn-induced cellular damage by inducing fibroblast activity and angiogenesis in porcine skin. These effects were also observed in humans. Chin et al. [19] showed that the blood of subjects administered with pearl powder had higher antioxidant activity compared to those of the placebo group. These findings suggest that pearl powder could be recommended for treating various age-related degenerative disorders.

On the other hand, we previously showed that the administration of nacre extract improved scopolamine-induced impairments, namely, short-term memory, object recognition and spatial memory [20]. Treatment with nacre extract restored the mRNA expression of brain-derived neurotrophic factor (BDNF), which was decreased after scopolamine treatment. These results suggest a new biological activity of nacre. In this study, we isolated and identified a bioactive component that improved scopolamine-induced memory impairment from nacre of pearl oyster shells and investigated the action mechanism of the bioactive substance.

## 2. Materials and Methods

### 2.1. Materials

Pearl oyster shells, *Pinctada fucata*, were obtained from Iki Bay, Nagasaki, Japan. Antibodies against β-actin, extracellular signal-regulated kinase (ERK), cAMP response element binding protein (CREB), phospho-ERK and phospho-CREB were purchased from Biorbyt (San Francisco, CA, USA). Scopolamine, trifluoroacetic acid (TFA), 5-bromo-4-chloro-3-indolyl phosphate, nitroblue tetrazolium, 2,2-diphenyl-1-picrylhydrazyl (DPPH), 2,4,6-tripyridyl-s-triazine and thiobarbituric acid were purchased from Fujifilm Wako (Osaka, Japan). An RNAiso Plus Kit was purchased from Takara (Shiga. Japan). An ABEE labeling kit plus S was purchased from J-Oil Mills (Tokyo, Japan). iTaq Universal SYBR Green supermix was purchased from Bio-Rad (Hercules, CA, USA). A BCA protein assay kit was purchased from Thermo Fisher Scientific (Kanagawa, Japan). A DEAE-5PW ion-exchange column and ODS reverse-phase column were purchased from Tosoh (Tokyo, Japan). Amicon Ultra was purchased from MercK Millipore (Waltham, MA, USA).

### 2.2. Preparation of the Nacre Extract

Nacre was obtained from the inner region of pearl oyster shells, crushed and ground to a powdered state, as described previously [20]. The nacre powder (~50 g) was decalcified in 5 L of 10% acetic acid for 1–2 weeks. The solution was dialyzed against deionized water, lyophilized, resuspended in deionized water and centrifuged. The resulting supernatant was the nacre extract.

### 2.3. Purification of the Bioactive Substance from the Nacre Extract

The nacre extract was fractionated using a 10 kDa Amicon Ultra. Fraction > 10 kDa was collected and subjected to a DEAE-5PW ion-exchange column equilibrated with 20 mM Tris-HCl (pH 7.5). Adsorbed components were eluted with an increasing linear concentration gradient of NaCl (0–0.5 M) and 20 mM Tris-HCl (pH 7.5). After the memory-improving activity of each fraction was investigated using the Y-maze test and novel object recognition test, the fraction containing memory-improving activity was pooled, dialyzed against deionized water and concentrated. The fraction was dissolved in 0.1% TFA solution, subjected to reverse-phase C18 column chromatography and eluted with a linear concentration gradient of acetonitrile from 0% to 50%. The fraction containing the memory-improving activity was pooled, concentrated and defined as nacre polysaccharide.

### 2.4. Animals

Four-week-old male ICR mice were purchased from CLEA Japan (Tokyo, Japan). Five or six mice were housed in a cage and maintained at 22 °C with free access to water and food. The mice were acclimatized for at least seven days and used for each experiment. Animal experiments were conducted following the guidelines of the Muroran Institute of Technology (approval number H29KS01), and approved by the Committee on the Ethics, Care, and Use of Animal Experiments of the Muroran Institute of Technology. Fractions from the nacre extract or nacre polysaccharides were administered intraperitoneally once a day for 14 consecutive days, according to the schedule shown in Figure 1; the mice in the control and scopolamine-treated groups were treated with the same volume of PBS or scopolamine (2 mg/kg, intraperitoneally) from day 8 to day 14. Mice were killed after the Y-maze test, and the novel object recognition test was performed on day 14. The brain was rapidly collected, and the cerebral cortex was excised and frozen at −80 °C until use.

### 2.5. Short-Term Memory Evaluation Using the Y-maze Test

The Y-maze test was performed as described previously [20,21]. Briefly, the mice were placed in the center of the Y-maze, and the sequence and number of arm entries were recorded for each mouse for 10 min. Alternation was defined as successive entries into arms, and spontaneous alternation was calculated using the following equation: spontaneous alternation (%) = [(number of alternations)/(total arm entries−2)] × 100. One hour before the test, mice were intraperitoneally administered with a fraction of the nacre extract or PBS. The dosages were determined based on the results obtained from a preliminary experiment or the ratio of each fraction obtained after column chromatography. After 30 min, memory impairment was induced by administering scopolamine (2 mg/kg), and the test was started 30 min later. The control group was administered with PBS only.

### 2.6. The Novel Object Recognition Test

The novel object recognition test was performed as described previously [20,21]. During the training session, the mice were placed in a test box in which two objects were introduced, and mice were allowed to explore each object for 5 min. The mice were then returned to their cages. After 24 h, one object was replaced by a novel object, and the mice were placed in the test box again. The relative exploration time (time spent exploring the replaced novel object × 100/total exploration time) was measured. One hour before the training session, mice were intraperitoneally administered with a fraction of the nacre extract or PBS. After 30 min, memory impairment was induced by injecting scopolamine (2 mg/kg), and the training session was started 30 min later. The control group was administered with PBS only.

### 2.7. Monosaccharide Composition Analysis

The monosaccharide composition of the isolated polysaccharide was analyzed using an ABEE Labeling Kit Plus S. The uronic acid content was determined using the modified sulfuric acid-carbazole method [22]. The sulfate content was determined by the barium chloride-gelatin turbidimetry method [23] using sulfuric acid as the standard. Protein and carbohydrate concentrations were determined using a BCA protein assay kit and the phenol-sulfate method, respectively.

### 2.8. SDS-PAGE and Western Bot Analysis

Western blotting was performed as described previously [24]. The cerebral cortex collected at the end of the animal experiment was homogenized, sample buffer containing 2% SDS and bromophenol were added and SDS polyacrylamide gel electrophoresis [25] was carried out. The proteins were transferred onto a polyvinylidene difluoride membrane and then incubated at room temperature for 2–6 h with 5% horse serum (*w/v*) in a solution containing 0.5 M NaCl, 20 mM Tris-HCl (pH 7.5) and 0.05% Tween 20 (solution A). The membrane was then incubated overnight with antibodies against β-actin, CREB, p-CREB, ERK and p-ERK. The membrane was treated for 2 h with an alkaline phosphatase-conjugated secondary antibody, and color development was then achieved using 5-bromo-4-chloro-3-indolyl phosphate and nitroblue tetrazolium. The band intensities were estimated using the ImageJ software.

### 2.9. Real-Time Polymerase Chain Reaction (PCR)

Total RNA from the cerebral cortex was extracted and purified using the RNAiso Plus Kit. After cDNA synthesis, real-time PCR was performed using primers specific for genes encoding GAPDH, BDNF, nerve growth factor (NGF), Mn-superoxide dismutase (SOD), Cu, Zn-SOD, catalase, interleukin (IL)-1β, tumor necrosis factor (TNF)-α and IL-6 (Table 1).

### 2.10. FT-IR ATR Spectroscopy

Infrared (IR) spectrometry (JASCOFT-IR 4000) of polysaccharides was performed in the 4000–400 cm^−1^ region to detect functional groups.

### 2.11. MALDI-TOF MS Analysis

MALDI-TOF MS analyses were performed using a Bruker (Bremen, Germany) UltrafleXtreme™ mass spectrometer. The nacre polysaccharide was partially hydrolyzed with 2 M TFA for 4 h at 80 °C. A mixture of TFA-treated polysaccharide and dihydroxybenzoic acid as the matrix was analyzed.

### 2.12. Antioxidative Activities

After the cerebral cortex was homogenized in a solution containing 20% sucrose, the supernatant (the cerebral cortex extract) was used to determine the antioxidant activity. For the DPPH radical scavenging activity, DPPH solution (0.8 mg/mL) in 50% ethanol was mixed with the cerebral cortex extract, and the decrease in absorbance was measured at 517 nm for 30 min [24].

Fe^3+^-reducing activity was measured according to the method described by Benzie and Strain [26]. A solution containing 250 mM acetate buffer (pH 3.6), 0.8 mM 2,4,6-tripyridyl-s-triazine, 1.6 mM FeCl_3_·6H_2_O and the cerebral cortex extract was mixed, and the absorbance was measured at 593 nm.

Lipid peroxidation was determined by measuring the malondialdehyde (MDA) content. The cerebral cortex extract was mixed with 50% trichloroacetic acid, and the mixture was centrifuged at 14,000× *g* for 10 min. The supernatant was incubated with 0.67% thiobarbituric acid at 100 °C for 10 min, and the absorbance at 540 nm was measured [24].

### 2.13. Statistical Analysis

Each experiment was performed at least two or three times. Data are expressed as the mean ± standard deviation (SD) of 5–6 mice per each group. One-way analysis of variance and the Tukey–Kramer multiple-comparison test were used to analyze the data using Excel Statistics software (SSRI, Tokyo, Japan). A *p* value < 0.05 was considered statistically significant.

## 3. Results

### 3.1. Isolation of a Substance That Improves Scopolamine-Induced Memory Impairment from the Nacre Extract

We previously showed the memory-improving activity of the nacre extract against scopolamine-induced memory impairment. In this study, we isolated a bioactive substance with memory-improving activity against scopolamine-induced memory impairment, which was assessed using the Y-maze and novel object recognition tests. To purify the bioactive substance, the nacre extract was first fractionated using 10 kDa Amicon Ultra. Each fraction > 10 and < 10 kDa was pooled, and the behavioral tests were performed. The Y-maze test revealed that scopolamine-treated mice showed a significant decrease in spontaneous alternation compared to those in the control group (Figure 1). Spontaneous alternation in Y-maze test is believed to reflect spatial working memory and short-term memory. The decrease was recovered by treatment with the nacre extract and >10 kDa fraction, although the difference was not significant. The memory-improving activity of the >10 kDa fraction was also observed in the novel object recognition test (Figure 2). As mice have a preference for novelty, if the mouse remembers the familiar object from the training session, it will spend most of its time at the new object. Scopolamine-treated mice spent less time exploring the new object (50%) compared to those in the control group (61%). On the other hand, mice injected with both scopolamine and the nacre extract or >10 kDa fraction significantly recovered the relative exploration time of the new object (58%). These findings indicated that the >10 kDa fraction can improve scopolamine-induced memory impairment. Next, the >10 kDa fraction was subjected to a DEAE-5PW ion-exchange column and the adsorbed components were eluted with a concentration gradient of 0 to 0.5 M NaCl (Figure 3). Four fractions were pooled, and the Y-maze and novel object recognition tests were performed. Using both Y-maze and novel object recognition tests, we found fraction 4 to show memory-improving activity in both tests (Figure 3). Fraction 4 was further purified using a C18 reverse-phase column. The absorbed components were eluted with an acetonitrile concentration gradient of 0% to 50%. Three fractions were pooled, and memory-improving activity was investigated. Memory-improving activity was observed in fraction 3, which was eluted at a higher concentration of acetonitrile (Figure 4). Fraction 3 was pooled and used to identify the memory-improving substance in the following experiment. The yield of the memory-improving substance was approximately 0.3 mg from 150 g of nacre powder. The isolated substance showed memory-improving activity at a dose of 50 μg/kg in contrast to 50 mg/kg in the nacre extract in the Y-maze and novel object recognition tests (Figure 5), indicating that fraction 3 is the memory-improving substance in the nacre extract.

### 3.2. Identification of the Memory-Improving Substance

The FT-IR spectra of isolated fraction 3 displayed a stretching intense characteristic peak at 1037 cm^−1^ for C-O-C and C-O-H of the pyranose ring [27] (Figure 6). The stretching peak at 1630 cm^−1^ suggested the presence of a carboxyl group [28]. In addition, the broad absorption at 1258 cm^−1^ was characteristic of the S-O stretching vibration of the sulfate group [29]. Next, TFA-treated fraction 3 was analyzed using MALDI-TOF MS (Figure 6). The spectrum showed a uniform and Gaussian distribution of peaks, which are often observed for polysaccharides [30,31,32]. Limited acid hydrolysis of the fraction 3 led to an accumulation of a peak of mass 486 Da, which corresponded to the tri-hexose unit, and 132 Da, which corresponded to pentose. These results demonstrate that isolated fraction 3 is a sulfate polysaccharide containing uronic acid named nacre polysaccharide.

Protein and sugar contents of the nacre polysaccharide were approximately 1% and 99%, respectively. When the nacre polysaccharide was resolved on a polyacrylamide gel, one band was observed by toluidine blue staining (Figure 6). Based on gel filtration analysis, the molecular weight was approximately 75 × 10^4^. Monosaccharide composition analysis showed that the nacre polysaccharide is a heteropolysaccharide, in that it is rich in glucose, galactose, mannose and ribose (Table 2). Sulfate and uronic acid contents of the nacre polysaccharide were 10.5% and 7.8%, respectively. These results also indicate that the nacre polysaccharide is a sulfate polysaccharide.

### 3.3. The Nacre Polysaccharide Reduces Oxidative Stress and Inflammation

To identify the action mechanism of the nacre polysaccharide, we determined the levels of oxidative stress in the brains of mice, because scopolamine induces oxidative stress and the production of reactive oxidant species. Scopolamine administration slightly increased lipid peroxidation levels compared to those in control mice (Figure 7). Treatment with nacre polysaccharides significantly reduced lipid peroxidation levels compared to those in scopolamine-treated mice. The DPPH radical scavenging activity and Fe^3+^-reducing activity in the cerebral cortex also increased after administration of the nacre polysaccharide (Figure 7), suggesting that this polysaccharide can reduce oxidative stress. To confirm this result, the mRNA expression levels of antioxidant enzymes were measured by real-time PCR. Treatment with scopolamine significantly reduced mRNA expression levels of Cu, Zn-SOD and catalase compared to those of the control group, and the administration of the nacre polysaccharide had a tendency to reverse this effect (Figure 7), denoting that nacre polysaccharides can improve scopolamine-induced oxidative stress. We further assessed the effects of nacre polysaccharides on the expression levels of inflammatory mediators (Figure 8). Scopolamine treatment significantly elevated the expression levels of the inflammatory cytokines IL-1β, IL-6 and TNF-α, whereas nacre polysaccharides restored these levels. Collectively, these findings suggest that nacre polysaccharides suppress scopolamine-induced inflammation.

### 3.4. Nacre Polysaccharides Increase mRNA Levels of Memory-Related Genes

To understand other mechanisms through which the nacre polysaccharide protects against scopolamine-induced memory impairment, the mRNA expression levels of neurotrophic factors that regulate neuroplasticity were examined (Figure 8). Although scopolamine markedly downregulated both NGF and BDNF mRNA expression levels, nacre polysaccharides significantly reversed this downregulation. These results show that nacre polysaccharides effectively recover the expression of two central neurotrophins that are involved in memory formation and storage. To further confirm this finding, we examined whether the nacre polysaccharide activates the ERK-CREB pathway that regulates the expression of NGF and BDNF using Western blot analysis (Figure 9). Although treatment with scopolamine did not decrease the phosphorylation levels of ERK and CREB, nacre polysaccharides significantly increased their phosphorylation levels, suggesting that nacre polysaccharides can activate the ERK-CREB pathway.

## 4. Discussion

In this study, we isolated the nacre polysaccharide that has memory-improving activity against scopolamine-induced memory impairment. The nacre polysaccharide is a uronic acid-containing sulfate polysaccharide. It is well known that the nacreous layer in pearl oyster shells contains chitin, which plays an important role in shell formation [33]. However, the existence of polysaccharides other than chitin remains obscure. Previous studies have demonstrated the involvement of polysaccharides in the nucleation of shell and the involvement of sulfates and carboxylates in the orientation of calcite nucleation [34,35]. Isolated nacre polysaccharides may be involved in the nucleation of shell.

Various kinds of polysaccharides reportedly have memory-improving activity against various animal models, such as those with amyloid b- and scopolamine-induced memory impairment [36,37,38,39]. It is generally believed that only some small molecules can cross the blood–brain barrier (BBB) via lipid mediation. However, some oligosaccharides can cross the BBB, such as chitosan, GM1, [40,41] and gastrodin (molecular weight of 4.7 kDa) [42] oligosaccharides. These results suggest that the metabolites of the nacre polysaccharide with a high molecular weight of 750 kDa may pass through the BBB and function in the brain. Therefore, it is necessary to investigate the metabolites and pharmacokinetics of nacre polysaccharides in the future.

Our results showed that the nacre polysaccharide significantly increased the phosphorylation levels of ERK and CREB. Many studies have shown that the activation of ERK-CREB signaling enhances cognitive function [43,44,45]. The memory-improving activity of nacre polysaccharides may also be due to the activation of ERK-CREB signaling. This is supported by an increase in BDNF and NGF mRNA expression levels, which are regulated by the activation of CREB [46,47]. Furthermore, nacre polysaccharide treatment restored the mRNA expression levels of BDNF and NGF, which were considerably decreased by scopolamine treatment. Houeland et al. [48] reported that NGF deprivation causes Alzheimer-like pathologies, such as amyloid-β accumulation, synaptic dysfunction and memory deficits, and NGF treatment can reverse these pathological changes. BDNF activates TrkB and enhances synaptic plasticity, memory formation and memory storage persistence [49,50]. Moreover, studies on animal models showed that an increase in BDNF and NGF expression is effective in improving memory impairment and depression. Thus, nacre polysaccharides may also be effective in ameliorating amyloid β-induced and d-galactose-induced memory impairment.

Furthermore, scopolamine leads to increased oxidative stress, elevated inflammation, and blocking of the acetylcholine receptor in the brain [51,52,53]. Oxidative stress causes a significant increase in MDA levels, which is an index of lipid peroxidation, and decreases the expression levels of antioxidant enzymes. In the present study, nacre polysaccharides decreased MDA levels and suppressed the scopolamine-induced decrease in the expression levels of antioxidant enzymes. In addition, nacre polysaccharides also attenuate neuroinflammation, which is linked to oxidative stress. Although scopolamine increased the expression of TNF-α, IL-1β and IL-6, which are inflammatory markers, nacre polysaccharides reduced the overexpression of these proteins. Oxidative stress and neuroinflammation have been reported to be associated with neurodegenerative diseases and aging [54,55,56]. Chiu et al. [19] reported that pearl powder has antioxidant activity and pearl powder-supplemented subjects showed an increase in antioxidant ability in blood. Hereby, these results show that nacre polysaccharides may be effective in treating neurodegenerative diseases associated with oxidation and inflammation.

## 5. Conclusions

Taken together, the present results show that the nacre polysaccharide recovers scopolamine-induced memory impairment by increasing the expression of BDNF and NGF and showing antioxidant and anti-inflammatory activities.

## Figures and Tables

**Figure 1 antioxidants-10-00505-f001:**
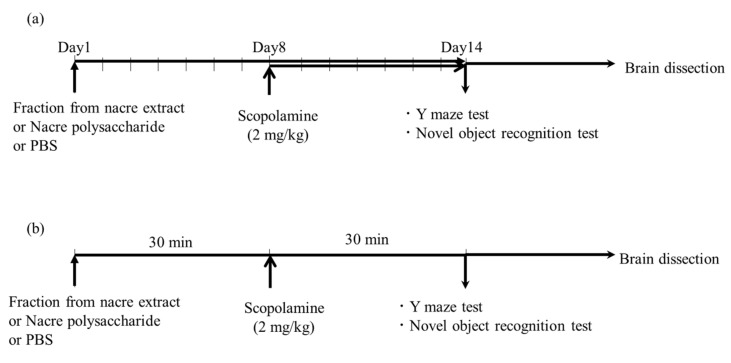
Schematic diagram of behavioral tests. (**a**) Mice were intraperitoneally injected with the nacre polysaccharide, nacre extract or PBS for 2 weeks. Scopolamine was injected for 1 week from day 8 to day 14, and behavioral tests were performed on day 14. (**b**) Experimental schedule of Y-maze test and novel object recognition test. One hour before conducting the behavioral tests, mice were intraperitoneally administered with a fraction of the nacre extract or PBS. After 30 min, memory impairment was induced by administering scopolamine (2 mg/kg), and the test was started 30 min later. The control group was administered with PBS only.

**Figure 2 antioxidants-10-00505-f002:**
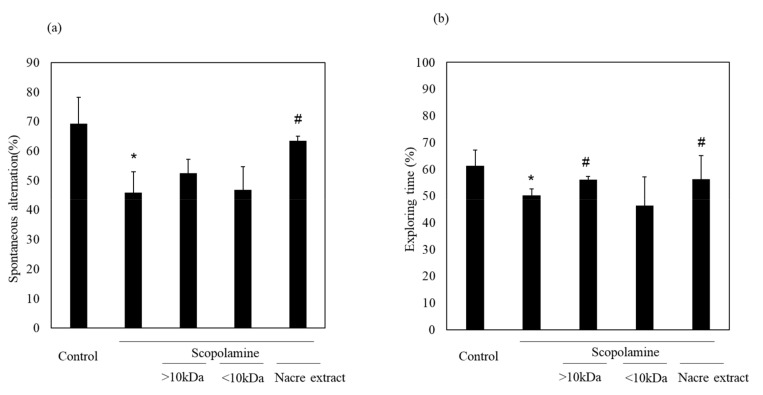
Effect of each fraction of >10 and <10 kDa on scopolamine-induced memory impairment. (**a**) Each fraction of the nacre extract and scopolamine was administered sequentially before conducting the behavioral tests: the Y-maze test (**a**) and novel object recognition test (**b**). Each fraction was injected at a dose of 5 mg/kg. Values are expressed as means ± SD. Significant difference between the groups: * *p* < 0.05, control versus scopolamine group; # *p* < 0.05, scopolamine group versus each fraction-treated group.

**Figure 3 antioxidants-10-00505-f003:**
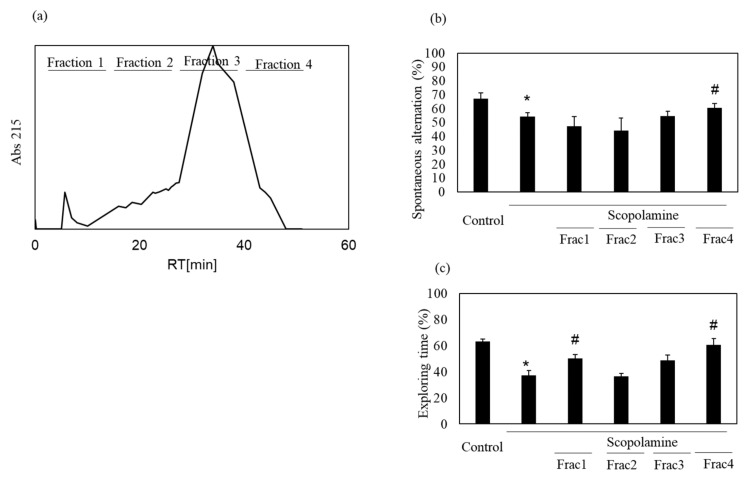
Effect of each fraction after DEAE-5PW ion-exchange column chromatography on scopolamine-induced memory impairment. (**a**) Fraction > 10 kDa was subjected to DEAE-5PW ion-exchange column chromatography. Adsorbed proteins were eluted with a 0–0.5 M concentration gradient of NaCl and four fractions were pooled. Memory-improving activity of each fraction was assessed by the Y-maze test (**b**) and novel object recognition test (**c**). Each fraction was administered at a dose of 0.18, 0.14, 2.7 and 0.80 mg/kg based on the amount of each fraction obtained after column chromatography. Values are expressed as means ± SD. Significant difference between the groups: * *p* < 0.05, control versus scopolamine group; # *p* < 0.05, scopolamine group versus each fraction-treated group.

**Figure 4 antioxidants-10-00505-f004:**
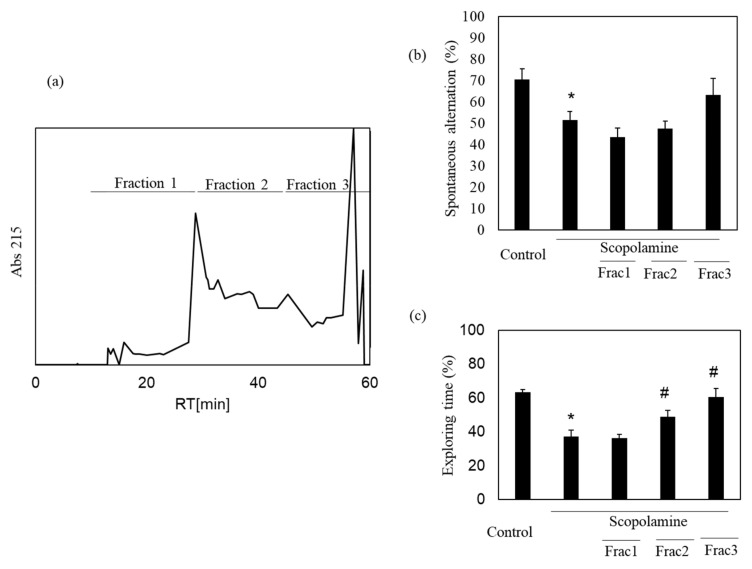
Effect of each fraction after C18 reverse-phase column chromatography on scopolamine-induced memory impairment. (**a**) Fraction 4 after DEAE-5PW ion-exchange column chromatography was subjected to C18 reverse-phase column chromatography and was eluted with a 0–50% acetonitrile concentration gradient. Three fractions were pooled and memory-improving activity of each fraction was measured by the Y-maze test (**b**) and novel object recognition test (**c**). Each fraction was administered at a dose of 0.03, 0.05 and 0.05 mg/kg. Values are expressed as means ± SD. Significant difference between the groups: * *p* < 0.05, control versus scopolamine group; # *p* < 0.05, scopolamine group versus each fraction-treated group.

**Figure 5 antioxidants-10-00505-f005:**
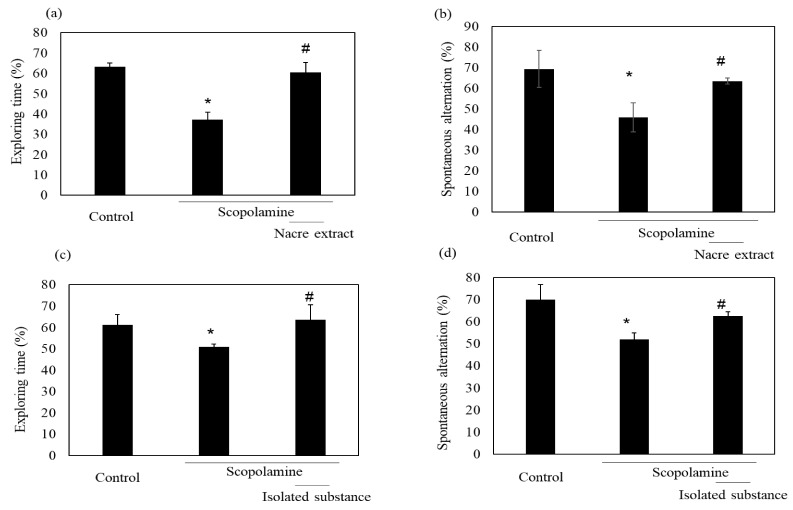
Comparison of the isolated substance and the nacre extract against scopolamine-induced memory impairment. The isolated substance of 50 μg/kg (**c**,**d**) or nacre extract of 50 mg/kg (**a**,**b**) and scopolamine were administered sequentially before conducting the behavioral tests, and spontaneous alternation (**b**,**d**) and relative exploring time (**a**,**c**) were determined. Values are expressed as means ± SD. Significant difference between the groups: * *p* < 0.05, control versus scopolamine group; # *p* < 0.05, scopolamine group versus nacre extract or isolated substance-treated group.

**Figure 6 antioxidants-10-00505-f006:**
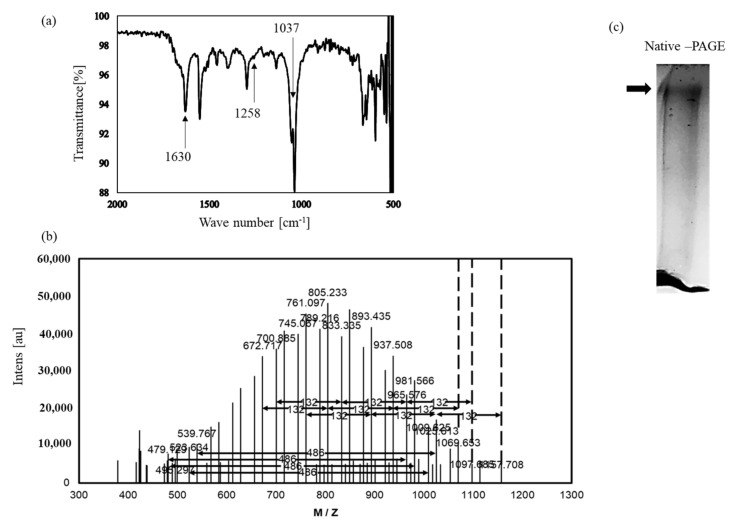
Identification of the isolated substance. (**a**) FT-IR ATR spectrum of the isolated substance. (**b**) MALDI-TOF MS spectrum of the isolated substance. (**c**) The isolated substance was separated by polyacrylamide gel electrophoresis and stained with toluidine blue.

**Figure 7 antioxidants-10-00505-f007:**
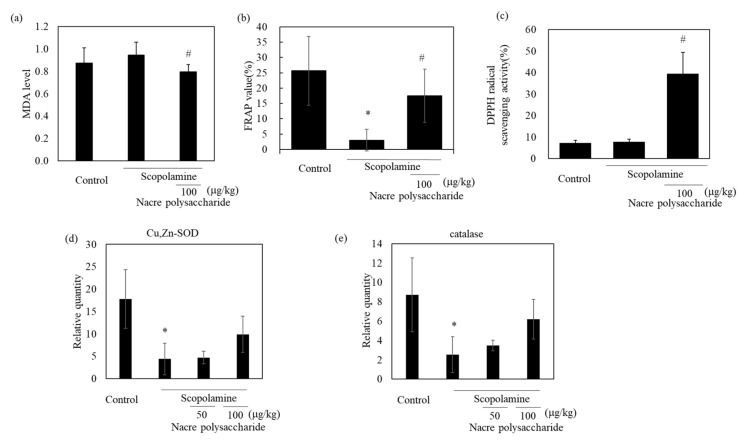
Effect of the nacre polysaccharide on oxidative stress in the cerebral cortex of scopolamine-treated mice. Lipid peroxidation (**a**) and antioxidant activities (Fe^3+^-reduction activity (**b**); DPPH radical scavenging activity (**c**)) in cerebral cortex of mice were determined. Expression levels of the antioxidant enzymes, Cu, Zn-SOD (**d**) and catalase (**e**), were examined by real-time PCR. The expression of each gene was normalized to the mean expression of GAPDH compared with the control values. Values are expressed as means ± SD. Statistical significance was determined using one-way analysis of variance with Tukey’s test. Significant difference between the groups: * *p* < 0.05, control versus scopolamine group; # *p* < 0.05, scopolamine group versus nacre polysaccharide group.

**Figure 8 antioxidants-10-00505-f008:**
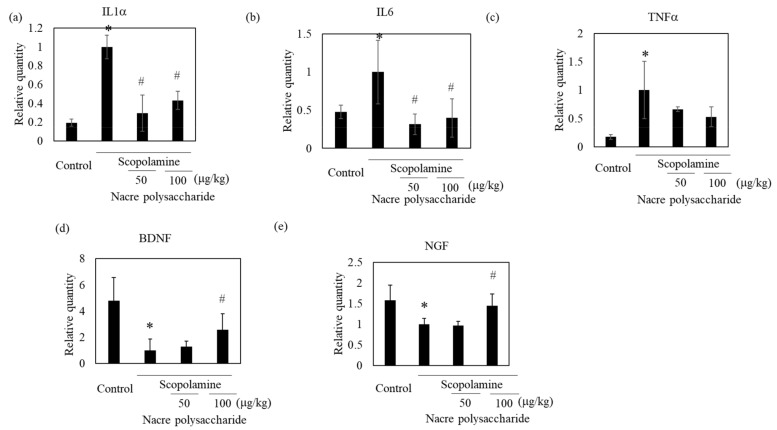
Effect of the nacre polysaccharide on the expression of genes associated with neuronal growth and inflammation in the cerebral cortex of scopolamine-treated mice. mRNA expression levels of IL-1β (**a**), IL-6 (**b**), TNF-α (**c**), BDNF (**d**) and NGF (**e**) were assessed by real-time PCR. The expression of each gene was normalized to the expression of GAPDH compared with control values. Values are expressed as means ± SD. Statistical significance was determined using one-way analysis of variance with Tukey’s test. Significant difference between the groups: * *p* < 0.05, control versus scopolamine group; # *p* < 0.05, scopolamine group versus nacre polysaccharide group.

**Figure 9 antioxidants-10-00505-f009:**
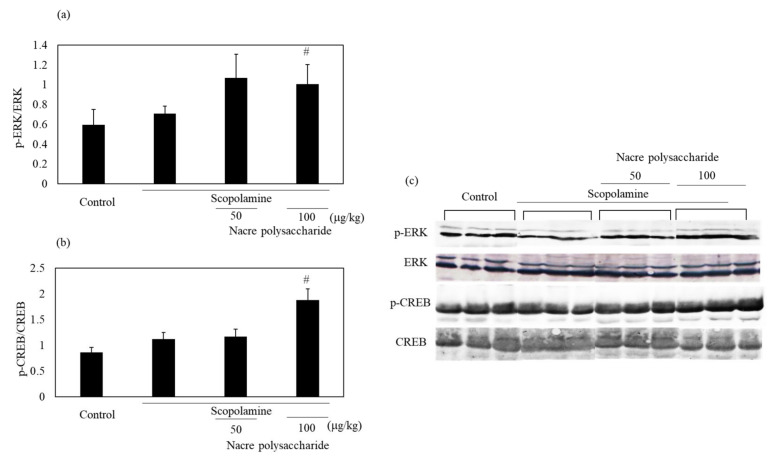
Effect of the nacre polysaccharide on the phosphorylation levels of ERK (**a**) and CREB (**b**) in the cerebral cortex of scopolamine-treated mice. The phosphorylation levels of ERK and CREB were determined by Western blotting (**c**). Values are expressed as means ± SD. Statistical significance was determined using one-way analysis of variance with Tukey’s test. Significant difference between the groups: # *p* < 0.05, scopolamine group versus nacre polysaccharide group.

**Table 1 antioxidants-10-00505-t001:** Primer sequence.

Gene Name	Accession Number	Primer	Sequence
Gapdh	BC023196.2	GAPDH-F	5′-TGACCTTGCCCACAGCCTTG-3′
		GAPDH-R	5′-CATCACCATCTTCCAGGAGCG-3′
Bdnf	EF125669.1	BDNF-F	5′-AGAGCTGCTGGATGAGGACCAG-3′
		BDNF-R	5′-CAAAGGCACTTGACTACTGAGCA-3′
Cas	NM_009804.2	Catalase-F	5′-AGGTGTTGAACGAGGAGGAG-3′
		Catalase-R	5′-TGCGTGTAGGTGTGAATTGC-3′
CuZnSOD	NM_011434.2	CuZn-SOD-F	5′-CGGATGAAGAGAGGCATGTT-3′
		CuZn-SOD-R	5′-CACCTTTGCCCAAGTCATCT-3′
IL-1beta	NM_008361.4	IL-1β-F	5′-GGGCCTCAAAGGAAAGAATC-3′
		IL-1β-R	5′-TACCAGTTGGGGAACTCTGC-3′
Il-6	X54542.1	IL-6-F	5′-AGACTTCCATCCAGTTGCCT-3′
		IL-6-R	5′-CAGGTCTGTTGGGAGTGGTA-3′
Ngf	V00836.1	NGF-F	5′-CAGTGTCAGTGTGTGGGTTG-3′
		NGF-R	5′-TGTGAGTCGTGGTGCAGTAT-3′
TNF-alpha	BC137720.1	TNF-α-F	5′-ACGGCATGGATCTCAAAGAC-3′
		TNF-α-R	5′-GTGGGTGAGGAGCACGTAGT-3′

**Table 2 antioxidants-10-00505-t002:** Sugar composition.

Monosaccharide	Composition [%]
D-Galactose	23.6
D-Mannose	12.0
D-Glucose	26.0
D-Ribose	5.6
N-Acetyl-D-Mannosamine	3.9
N-Acetyl-D-Glucosamine	6.8
Fucose	3.8
Rhamnose	7.9
N-Acetyl-D-Galactosamine	2.6
Uronic Acid	7.8
Sulfate Group	10.5

## Data Availability

All data included in this study are available upon request by contacting the corresponding author.

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
