# Peer review of "Sulfated Polysaccharide Isolated from the Nacre of Pearl Oyster Improves Scopolamine-Induced Memory Impairment"

_antioxidants, 2021, doi:10.3390/antiox10040505_

Round 1
Reviewer 1 Report
This manuscript by Yamagami and collaborators entitled “Sulfated polysaccharide isolated from the nacre of pearl oyster 2 improves scopolamine-induced memory impairment” focuses on the characterisation of a pearl oyster nacre extract and its effects on memory. Although the work is potentially interesting, it is too preliminary and requires additional experiments and control to be considered suitable for publication.
- The figures are too small and difficult to assess. Authors need to make them bigger and at higher resolution
-Authors should provide additional information regarding number of replicas in their experiments
-How many control mice did the authors used?
-Why did the authors only select male mice for their experiments? Does this introduce a bias in their experiments?
-Behavioural tests need to be clearly represented in the paper
-What is the error in the Y-maze experiment? Is that difference significant?
-The representation of the statistical analysis in difficult to read
-I am also not convinced that all differences are significant (particularly in Figure 7)
-I cannot assess the blot in Figure 9c. The quality of the picture is too low, and the contrast is too high
-The text in the figures is too small and difficult to read
Author Response
We wish to express our strong appreciation to the reviewers for their insightful comments on our paper. We feel the comments have helped us significantly improve the paper. We attach here our revised manuscript and point-by-point response to the reviewer’s comments.
- The figures are too small and difficult to assess. Authors need to make them bigger and at higher resolution
In accordance with reviewer’s comment, we have changed all figures.
-Authors should provide additional information regarding number of replicas in their experiments
In accordance with reviewer’s request, we have added the following text. “Each experiment was performed at least two or three times.” (L201)
-How many control mice did the authors used?
We used 5-6 mice per each group in all experiments. The sentence was added in Materials and method section (L202).
-Why did the authors only select male mice for their experiments? Does this introduce a bias in their experiments?
Many researchers use male mice in behavioral studies, because estorous cycles of female mice may cause data variability.
-Behavioural tests need to be clearly represented in the paper
In accordance with reviewer’s request, we have added the text in the Result section (L218-223)
-What is the error in the Y-maze experiment? Is that difference significant?
The variation into the data is due to individual differences among mice.
Therefore, two or three independent experiments were carried out in all experiments.
Although significant difference was not detected in Y-maze test between scopolamine group and each fraction-treated group because of large standard deviation (p=0.068), the tendency to reverse scopolamine-induced decrease was observed in fraction 3 (Fig. 4).
-The representation of the statistical analysis in difficult to read
-I am also not convinced that all differences are significant (particularly in Figure 7)
We appreciate you pointing out our mistakes. We have confirmed all data again and performed statistical analysis using Excel Statistics software (SSRI, Tokyo, Japan). Data are expressed as the mean ± standard deviation (SD) of 5-6 mice per group. Several experiments were carried out again and revised. Significant differences between the groups (control versus scopolamine group, scopolamine group versus nacre polysaccharide or extract group) were estimated using Excel Statistics software and some data has been corrected.
We could not detect significant differences between scopolamine group and nacre polysaccharide group in (e) and (f) in Fig. 7. However, the administration of the nacre polysaccharide had a tendency to reverse scopolamine-induced decrease in several experiments (L310-312).
-I cannot assess the blot in Figure 9c. The quality of the picture is too low, and the contrast is too high
In accordance with reviewer’s request, we have changed the figure in Fig 9.
-The text in the figures is too small and difficult to read
In accordance with reviewer’s comment, we have changed all figures.
Thank you again for your comments on our paper. We trust that the revised manuscript is suitable for publication.
Reviewer 2 Report
The paper prepared by Yamagami et al. is very interesting and provide new information in the current research area. I have some comments which need to be included before publication:
- In the abstract, please add one-two sentences about the introduction to the study subject
- Some editorial bugs, please correct it
- Lines 31 and 35, please add the proper reference
- In the Materials section, please provide all used reagents and materials.
- Please unify the manufacturer description
- The analysis of polysaccharides description should be presented before biochemical analysis
- The statistical analysis is not correctly performed. Why Authors use SEM? – SD should be presented. How normality was established, how equality of variance was evaluated? – this is necessary for the TUKEY test.
- Number of Bioethics commission agreement must be provided
Author Response
We wish to express our strong appreciation to the reviewers for their insightful comments on our paper. We feel the comments have helped us significantly improve the paper. We attach here our revised manuscript and point-by-point response to the reviewer’s comments.
- In the abstract, please add one-two sentences about the introduction to the study subject
In accordance with reviewer’s comment, we have added the text (L10-11) in the abstract.
- Some editorial bugs, please correct it.
We have corrected editorial bugs.
- Lines 31 and 35, please add the proper reference
In accordance with reviewer’s comment, we have added references (L32 and L38).
- In the Materials section, please provide all used reagents and materials.
Please unify the manufacturer description
In accordance with reviewer’s comment, we have added reagents and materials in the Materials section and unified the manufacturer description (L69-77).
- The analysis of polysaccharides description should be presented before biochemical analysis
In accordance with reviewer’s comment, we have described FT-IR and MALDI analysis of nacre polysaccharide before biochemical analysis (L297-320).
- The statistical analysis is not correctly performed. Why Authors use SEM? – SD should be presented. How normality was established, how equality of variance was evaluated? – this is necessary for the TUKEY test.
We appreciate you pointing out our mistakes. We have confirmed all data again and performed statistical analysis using Excel Statistics software (SSRI, Tokyo, Japan). Data are expressed as the mean ± standard deviation (SD) of 5-6 mice per group. Several experiments were carried out again and revised. Significant differences between the groups (control versus scopolamine group, scopolamine group versus nacre polysaccharide or extract group) were estimated using Excel Statistics software and some data has been corrected.

- Number of Bioethics commission agreement must be provided
We have added number of bioethics agreement (L105).
Thank you again for your comments on our paper. We trust that the revised manuscript is suitable for publication.
Round 2
Reviewer 1 Report
The paper significantly improved and it is suitable for publication.
Reviewer 2 Report
All my suggestions have been included in current version of manuscript. Thank You.